# Input-Adaptive Proxy for Black Carbon as a Virtual Sensor

**DOI:** 10.3390/s20010182

**Published:** 2019-12-28

**Authors:** Pak Lun Fung, Martha A. Zaidan, Salla Sillanpää, Anu Kousa, Jarkko V. Niemi, Hilkka Timonen, Joel Kuula, Erkka Saukko, Krista Luoma, Tuukka Petäjä, Sasu Tarkoma, Markku Kulmala, Tareq Hussein

**Affiliations:** 1Institute for Atmospheric and Earth System Research (INAR)/Physics, University of Helsinki, FI-00560 Helsinki, Finland; martha.zaidan@helsinki.fi (M.A.Z.); salla.sillanpaa@helsinki.fi (S.S.); krista.q.luoma@helsinki.fi (K.L.); tuukka.petaja@helsinki.fi (T.P.); markku.kulmala@helsinki.fi (M.K.); 2Helsinki Region Environmental Services Authority (HSY), P.O. Box 100, FI-00066 Helsinki, Finland; anu.kousa@hsy.fi (A.K.); jarkko.niemi@hsy.fi (J.V.N.); 3Atmospheric Composition Research, Finnish Meteorological Institute, FI-00560 Helsinki, Finland; hilkka.timonen@fmi.fi (H.T.); joel.kuula@fmi.fi (J.K.); 4Pegasor Oy, FI-33100 Tampere, Finland; erkka.saukko@pegasor.fi; 5Department of Computer Science, University of Helsinki, FI-00560 Helsinki, Finland; sasu.tarkoma@helsinki.fi; 6Department of Physics, The University of Jordan, Amman 11942, Jordan

**Keywords:** input-adaptive proxy, black carbon, robust linear regression, air quality, street canyon, urban background, virtual sensor

## Abstract

Missing data has been a challenge in air quality measurement. In this study, we develop an input-adaptive proxy, which selects input variables of other air quality variables based on their correlation coefficients with the output variable. The proxy uses ordinary least squares regression model with robust optimization and limits the input variables to a maximum of three to avoid overfitting. The adaptive proxy learns from the data set and generates the best model evaluated by adjusted coefficient of determination (adjR^2^). In case of missing data in the input variables, the proposed adaptive proxy then uses the second-best model until all the missing data gaps are filled up. We estimated black carbon (BC) concentration by using the input-adaptive proxy in two sites in Helsinki, which respectively represent street canyon and urban background scenario, as a case study. Accumulation mode, traffic counts, nitrogen dioxide and lung deposited surface area are found as input variables in models with the top rank. In contrast to traditional proxy, which gives 20–80% of data, the input-adaptive proxy manages to give full continuous BC estimation. The newly developed adaptive proxy also gives generally accurate BC (street canyon: adjR^2^ = 0.86–0.94; urban background: adjR^2^ = 0.74–0.91) depending on different seasons and day of the week. Due to its flexibility and reliability, the adaptive proxy can be further extend to estimate other air quality parameters. It can also act as an air quality virtual sensor in support with on-site measurements in the future.

## 1. Introduction

According to a report by the Health Effect Institute in 2019 [1], nearly one out of every ten worldwide deaths resulted from exposure to air pollution which is strongly associated with different cardiovascular and respiratory diseases. Air pollution costs heavily for the global economy and recent estimates suggest that 2–5% of gross domestic profit (GDP) is spent on treating diseases linked to air pollution according to World Health Organization (WHO) [2]. Currently, about 90% of the world’s population lives in urban areas where air pollution concentration exceeds safe limits [2].

In order to understand the nature of urban air pollution, continuous and reliable measurements of air pollutants are needed [3]. However, missing data remains one of the major challenges for air quality studies. The most common reasons for incomplete data sets include instrument failure, data corruption or human error in data acquisition [4,5]. Missing data can also be attributed to complicated, bulky and labour-intensive air analyzers [5]. Rubin [6] classified incomplete data according to their generating mechanisms, where air quality data sets are generally missing at random (MAR), which implies that the probability that a value is missing does not depend on the missing value [4]. The process of replacing missing data with substituted values is called imputation in statistics. Under MAR mechanism, the complete case analysis no longer relies on a random sample of the source population and selection of imputation methods tends to result in bias [7]. In time series analysis, this problem can be intensified because of the exclusion of incomplete observations, which may corrupt temporal structures such as autocorrelation, trends, and seasonality [8].

Currently, the easiest and the simplest way is imputation by median, but this tends to distort the marginal distribution of the data due to the higher frequency of observations around the average after imputation. Another common practice is univariate imputation by nearest neighbour, linear and cubic interpolation [4,9,10], where endpoints of the gaps are used as estimates for all missing values. It is straightforward but neither applicable nor reliable for data sets with long gaps. A more accurate way is multivariate imputation by unconditional mean by using the information from other measured covariates of the same study unit to impute the missing value based on the prediction from a linear regression model, as known as proximity estimation, hereafter as proxy [11]. This method works well when the variables used to sort the data are highly predictive of the variable with the missing values [12]. However, this method does not necessarily impute all data points if there are missing data gaps in the predictor variables indicated in the proxy. Instead of single imputation, multiple imputation estimation may consider additional variability and obtain confidence intervals, which is more precise. This has been advocated as a statistically sound approach [13], but so far its use has been limited mainly to the social and medical sciences [9]. A more sophisticated method is Bayesian proxy [10,14,15,16], but its performance is arguable. Junninen et al. [4] suggested that the Bayesian method may give less accurate results while Zaidan et al. [17] illustrated how Bayesian method prevents over-fitting naturally. More advanced yet computationally costly machine learning techniques, such as artificial neural network (for example multi-layer back-propagation nets and self-organizing maps) and support vector machine, tend to be black-box models where the relationship between variables and output are not transparent [5,17,18].

In recent years, the rapid development of reference stations [19] and low-cost sensor networks [20] generate massive amount of air quality data. It is crucial to develop a method to quickly impute all missing data with a reasonable and reliable approximation method. Furthermore, the variable, in which data gap was fully filled, can be considered to be used as a virtual sensor, which uses information from other available measurements to estimate the quantity of interest. The deployment of this concept will serve as a feasible and economical alternative to expensive or impractical physical measurement instruments [21].

The purpose of this paper is to modify the traditional multiple imputation method by simple regression proxy (hereafter as traditional proxy) and to evaluate the modified method (hereafter as input-adaptive proxy) by testing both methods on two black carbon (BC) data sets in Helsinki, Finland. Additionally, we investigate the possibility of utilizing input-adaptive proxy as virtual sensors in terms of its accuracy and coverage. Section 2 outlines the methods of the input-adaptive proxy. We describe the specification of sites and instruments and the data pre-processing procedures in Section 3. In Section 4, we briefly describe the seasonal and diurnal pattern of BC concentrations at two measurement sites, focusing on the performance of the proxies in different conditions. Finally, the possibility of turning the input-adaptive proxy into virtual sensors and future steps are discussed in Section 5.

## 2. Methods

This section describes the traditional multiple imputation method and our input-adaptive proxy modified on the foundation of traditional proxy. The workflow for the modified proxy development is shown in Figure 1. Section 2.1 describes the development of the input-adaptive proxy. The criteria of rejecting models and the grading system of the models are explained in detail in Section 2.2 and Section 2.3, respectively.

### 2.1. Proxy Development

The first step of exploring a given data set is to examine the number of relevant variables and its completeness. We then examine the type of distribution for each variable and transform the data when necessary to build a normal distribution for regression techniques in later stage. The most common methods include logarithmic transformation, inverse transformation and square transformation [19]. We calculate the absolute value of Pearson correlation coefficient (R) to investigate the linear correlation between the response variable and other explained variables by [22]:(1)$$R_{xy}=\frac{\sum_{i=1}^{N}\left(x_{i}-\bar{x}\right)\left(y_{i}-\bar{y}\right)}{\sqrt{\sum_{i=1}^{N}{\left(x_{i}-\bar{x}\right)}^{2}}\sqrt{\sum_{i=1}^{N}{\left(y_{i}-\bar{y}\right)}^{2}}}$$
where $\bar{x}$ is the arithmetic mean of the output variable and $\bar{y}$ is the mean of the each input variable. N is the number of valid data points in the variables of x and y. The R values range from 0 to 1, where 1 indicates the highest degree of correlation. We select input variables with high or moderate (R > 0.1) for the next step of model creation. This procedure is able to lower the number of models and to minimize the computational time. After that, we limit the number of input variables to a maximum of three. Although adding more input variables in the model can usually archive higher coefficient of determination, it increases the chance to get a worse prediction by overfitting. More input variables are also prone to incalculable estimations due to the possibility of data insufficiency [5]. 

We perform ordinary least squares (OLS) linear regression for each model of maximum three input variables generated in the previous step. In case of a model with p input variables, the generalized OLS regression model is [10,23,24]:(2)$$Y=\;\beta_{0}+\sum_{i=1}^{p}\beta_{i}X_{i}+\varepsilon$$
where Y is the output variable, $\beta_{0}$ is the intercept of the model, $\beta_{i}$ and $X_{i}$ correspond to the ith regression coefficient and ith input variable of the model, and ε is the random error with expectation 0 and variance *σ*².

We apply an extra regularization by using Tukey’s bisquare weighting function $\rho^{\prime }$ [25]:(3)$$\rho^{\prime }\left(r_{i}\right)=\{\begin{array}{c} r_{i}{\left\{1-\frac{r_{i}{}^{2}}{c}\right\}}^{2},\;\left|r_{i}\right|\leq c\\ \;0,\;\left|r_{i}\right|>c \end{array}$$
which $r_{i}$ is a function dependent on residuals, leverages from OLS fits and the estimates of the standard deviation of the error terms. A tuning factor c, equal to 4.685, is used [26]. This robust fitting is designed to be not overly affected by violations of OLS assumptions [27], and hence serves as an alternative to the traditional OLS regression [28] especially when data are contaminated with outliers that often take place in field measurements. We run the regression for every model and obtain its evaluation attributes (see more in Section 2.3). According to the attributes, we rank the models from the best performing one to the worst. The traditional proxy predicts the missing data with the estimated intercepts and coefficients only with the best performing model. Nevertheless, since data set in practice is typically incomplete, some data points in the input variables in the best model can possibly be missing, hence the imputation cannot be fully achieved. The input-adaptive proxy further imputes the missing data with the second best performing model, and so on, until all the voids are filled up. 

To enhance the reliability, we propose to use multiple imputation instead of single by the bootstrapping method [4,19,29]. Original data set is divided into five subsets each including random 80% of the data set. Separate robust OLS fittings are carried out to each subset. The final regression coefficients and evaluation attributions are computed by calculating the arithmetic mean of the resulting parameters in the five subsets [19]. Concerning the choice of the number of imputations, some used hundreds of data subsets, but it demands hundreds folds of computational time. It has been suggested that three to five imputations are sufficient to obtain excellent results [13]. However, when the percentage of missing data is larger, more imputations may be needed [30].

### 2.2. Model Rejection

Since the models are established by all possible combination of all selected features, some models might include interdependent variables, which might influence the resulting estimation. In atmospheric science field, air pollutants often interact and are dependent with each other. This interdependency is called multicollinearity in statistics [31].

When multicollinearity exists, the variances of the estimated regression coefficients for the independent variables are inflated. The variance inflation factor (VIF) for an estimated regression coefficient $\beta_{p}$ of a particular variable $X_{p}$ is a factor by which the variance is inflated. This quantifies the severity of multicollinearity in OLS regression analysis. The variance of OLS estimator for a typical regression coefficient $\beta_{p}$ and $VIF_{p}$ are given by the following equations, respectively [24,31]:(4)$$var\left(\hat{\beta_{p}}\right)=\frac{\sigma^{2}}{\sum_{i=1}^{N}{\left(X_{ip}-\bar{X_{p}}\right)}^{2}\left(1-R_{p}^{2}\right)}$$
(5)$$VIF_{p}=\frac{1}{1-R_{p}^{2}}$$
where $\sigma^{2}$ is the residual variance and $R_{p}^{2}$ is the unadjusted coefficient of determination when $X_{p}$ is regressed against all the other input variables in the model, and represents the proportion of variance of that variable that is associated with the other independent variables in the model. If there is no linear relation between $X_{p}$ and the other input variables in the model, $R_{p}^{2}$ will be zero and the variance of $\hat{\beta_{p}}$ will be $\sigma^{2}$/$\sum_{i=1}^{N}{\left(X_{ip}-\bar{X_{p}}\right)}^{2}$. By dividing this term, $VIF_{p}$ is obtained by (5). It is evident that a higher $VIF_{p}$ value leads to a higher proportion of variance for the $X_{p}$ input, which is related to the other input variables in the model. This means that severe multicollinearity effects are present. Kleinbaum et al. [31] suggested that the effect of multicollinearity can be ignored if the VIF is less than 10. Here, as well as a previous study by Fernández-Guisuraga et al. in 2016 [24], we use VIF = 5 as the threshold for model rejection. 

In addition, non-normality in the residuals and heteroscedasticity imply that the amount of error in the model is not consistent across the full range of the observed data. This violates the assumption of using OLS regression. Therefore, we use Lilliefors test to examine whether residuals from the model come from normally distributed population, and reject the model if the null hypothesis is not statistically significant (*p* < 0.05) [32,33]. It is a modification of the Kolmogorov-Smirnov (K-S) test, which corrects the K-S test for small values at the tails of probability distributions equation needed by calculating z-scores and test statistics based on the empirical distribution function of the z-scores for every single sample member [32,33].

### 2.3. Evaluation Attributes

In order to evaluate and rank the models, adjusted coefficient of determination ($adjR^{2}$), followed by mean absolute error (MAE) and root mean square error (RMSE) are used as diagnostic evaluation attributes. $adjR^{2}$ illustrates the linear association between two variables (the estimated output variable by proxy and the measured variable). This attribute also considers the degree of freedom, and adjusts the number of input terms in a model relative to the number of data points. We use $adjR^{2}$ over $R^{2}$ because $adjR^{2}$ takes the number of output data points into consideration [24]. In case of similar $adjR^{2}$ performance, we prefer to use the model with higher number of data points input to lower its uncertainty [22]:(6)$$adjR^{2}=1-\left(1-{\left(\frac{\sum_{i=1}^{N}\left(y_{i}-\bar{y}\right)\left(\hat{y_{i}}-\tilde{y}\right)}{\sqrt{\sum_{i=1}^{N}{\left(y_{i}-\bar{y}\right)}^{2}}\sqrt{\sum_{i=1}^{N}{\left(\hat{y_{i}}-\tilde{y}\right)}^{2}}}\right)}^{2}\right)\times \left(\frac{N-1}{N-p-1}\right)$$
where N is the number of complete data input to the model, p is the number of input variables, $y_{i}$ and $\hat{y_{i}}$ are ith measured and ith estimated output variable by the model, respectively, and $\bar{y}$ and $\tilde{y}$ is expected value of the measured valuable and the estimated output, respectively. However, $adjR^{2}$ does not consider the biases in the estimation. Therefore, we further validate the models with MAE and RMSE, where MAE measures the arithmetic mean of the absolute differences between the members of each pair, whilst RMSE calculates the square root of the average squared difference between the forecast and the observation pairs. RMSE is more sensitive to larger errors than MAE [5,24]:(7)$$MAE=\frac{1}{N}\sum_{i=1}^{N}\left|y_{i}-\hat{y_{i}}\right|$$
(8)$$RMSE=\;\sqrt{\frac{1}{N}\sum_{i=1}^{N}{\left(y_{i}-{\hat{y}}_{i}\right)}^{2}}$$
where N is the number of complete data input to the model, and $y_{i}$ and $\hat{y_{i}}$ are ith measured and ith estimated response variable by the model, respectively. Models will be first evaluated by $adjR^{2}$ to see how well the estimations fit with the original data with the consideration of degree of freedom. If $adjR^{2}$ have the same score, those models will be ranked by MAE and then RMSE on the basis of absolute error. Apart from ranking, *p*-values are also checked for each model in order to investigate whether the model is statistically significant. Standard error (SE) and classical *p*-values are also calculated to examine the significance of estimates of regression coefficients (β).

## 3. Case Studies: BC in Street Canyon and Urban Background

In this section, we present the evaluation of our input-adaptive proxy for two real data sets in Helsinki urban region. These two stations represent two different environments and they have different degree of missing data pattern. We chose BC as the variable to be filled as a case study. BC has a range of fine-size particulate matter (PM) and it is emitted as a by-product of combustion. In urban regions, BC is mainly originated from diesel-powered vehicles, wood combustion and long-range transport [34,35]. According to a recent report by WHO [36], BC might not be directly toxic, but it can act as a universal carrier of other chemical components with varying toxicity which can bring severe effects on human health, for example cardiopulmonary, respiratory diseases and diseases that are not related to allergies [37]. BC not only plays an important role in climate change by changing surface albedo when deposited and absorbing solar radiation [38], but also contributes to visibility and local air quality [39]. It has been recommended to include BC alongside with the other air quality parameters in air quality index because BC concentration can associate better with health effects of aerosol particles than just PM, which does not solely originate from combustion sources [36]. Nevertheless, continuous BC measurements are not necessarily carried out in every measurement stations. Therefore, BC is a good case study to deploy the modified proxy as a virtual sensor.

### 3.1. Site Description

Helsinki (60°10′ N, 24°56′ E) is the capital of Finland, located on a relatively flat land at the coast of Gulf of Finland. The area of the city is 715 km^2^ with about 650,000 inhabitants [40]. The two measurement stations are located at Mäkelänkatu and Kumpula in Helsinki, approximately 1 km from each other (Figure 2). They represent environments of street canyon and urban background, respectively. The climate type in Helsinki can be classified as continental or maritime based on the air flows and the pressure system [19].

Mäkelänkatu urabn supersite measurement station (Mäkelänkatu 50) is operated by the Helsinki Region Environmental Services Authority (HSY). The station is located in a street canyon in the immediate vicinity to one of the main streets of the city, 3 km from the Helsinki centre [41]. The street is one of the main roads leading to the city centre and it consists of six lanes and two tramlines. The annual mean traffic volume is 28,000 vehicles per workday. The traffic loads are especially high during rush hours at 8 in the morning and 5 in the afternoon. The buildings on both sides weaken the dilution process of pollutants. The supersite measurement station consists of a container (length 8.0 m, width 1.7 m, height 2.7 m), which is equipped with standard air quality measurement devices and other instruments. All the inlets for the measuring devices are located on the top of the container approximately at a height of 2.8 m from the ground level [42]. 

The Station for Measuring Ecosystem-Atmosphere Relations in Kumpula (SMEAR III) is situated on a rocky hill at 26 m above sea level, about 4 km northeast from the Helsinki centre. The surroundings of this urban background station are heterogeneous, constituting of buildings, small roads, parking lots, patchy forest and low vegetation. Residential activity, including wood combustion, accounts for a large portion of atmospheric pollutant source [19]. Trace gases are measured at a 31-m-high triangular lattice tower. Next to the tower stands a measurement container, where aerosols measurement instrumentation is located. In addition, basic meteorological measurements are made from the rooftop of Physicum building in the University of Helsinki. 

Apart from the trace gases, aerosols and meteorological measurements, traffic rates in Helsinki metropolitan area are monitored by the City of Helsinki. The nearest continuous counter station is in proximity to Mäkelänrinne swimming centre, about 600 m south and 1 km north from Kumpula and Mäkelänkatu measurement sites, respectively. The traffic monitoring does not take into account the split between light- and heavy-duty vehicles. It logs number of vehicles along Mäkelänkatu but it is estimated that 30% of traffic may have been diverted to Kumpulankatu before reaching Mäkelänkatu supersite. The actual positions are shown in Figure 2.

### 3.2. Instruments

BC mass concentration was measured by a multi-angle absorption photometer (MAAP) Thermo Scientific 5012 with a PM_1_ inlet at a 1-minute time resolution [43]. The MAAP determines the light absorbance from particles deposited on the filter using measurements of both transmittance and reflectance at different angles. The absorbance was converted to BC mass concentration by using a fixed 6.6 m^2^ g^−1^ mass absorption coefficient at wavelength of 637 nm [44].

Lung deposited surface area (LDSA), which is considered as a relevant metric for the negative health effects of aerosol particles [45], were collected with diffusion charging-based Pegasor AQ Urban sensor [46]. Particle mass concentration of diameter less than 2.5 µm (PM_2.5_) and less than 10 µm (PM_10_) were measured with continuous ambient particulate monitor TEOM 1405 in Mäkelänkatu and TEOM 1405-D in Kumpula [19]. Aerosol size distribution was also measured with a differential mobility particle sizer (DMPS) in combination of a differential mobility analyser (DMA) and a condensation particle counter (CPC). Mäkelänkatu used Vienna DMA and Airmodus A20 CPC while Kumpula used Twin DMPS (Hauke-type DMA and TSI Model 3025 CPC + Hauke-type DMA and TSI Model 3010 CPC) [47]. The DMPS systems measured particles with diameter 6–800 nm and 3–950 nm, respectively. The DMPS technique is based on the bipolar charging of aerosol particles, followed by classification of particles into size classes according to their electrical mobility (detailed procedures of data treatment of DMPS in [19]).

Nitrogen oxide (NO), nitrogen dioxide (NO_2_) and their sum NO_x_ in Mäkelänkatu and Kumpula were measured with chemiluminescence analyzers, Horiba APNA-370 and Thermo TEI42S, respectively. Ozone (O_3_) concentration was taken with UV photometric analyzers Horiba APOA-370 and Thermo Model 49i in Mäkelänkatu and IR-absorption photometer TEI49 in Kumpula. The measurement of carbon monoxide (CO) was collected with non-dispersive IR-absorption analyzer Horiba APMA-360 in Mäkelänkatu and Horiba APMA-370 in Kumpula. Sulphur dioxide (SO_2_) was only measured in Kumpula with a UV-fluorescence analyzer Horiba APSA-360.

All the meteorological variables, except radiation, in Mäkelänkatu were measured with Vaisala WXT 520 and Vaisala WXT536 weather transmitters. In Kumpula, horizontal wind speed and direction were measured by a Vaisala cup anemometer. Air temperature was measured with a platinum resistant thermometer Pt-100 and relative humidity (RH) with a platimun resistance thermometer and thin film polymer sensor Vaisala DPA500 on the roof of the University of Helsinki building. On the roof, air pressure was measured with a barometer Vaisala DPA500. Global radiation and photosynthetically active radiation (PAR) were also measured at the same place with a net radiometer and photodiode sensor Kipp and Zonen CNR1+PAR lite, respectively. Since there are no radiation measurements in Mäkelänkatu, we used the measurements of global radiation and PAR from Kumpula for the regression step. A list of the measured parameters and their instruments in Mäkelänkatu and Kumpula is demonstrated in Table 1.

### 3.3. Data Pre-Processing

In this study, we retrieved continuous data in the period between 1 January 2017 and 31 December 2018. This equals 730 days in total, which covers all four seasons. Traffic data were logged at 1-hour intervals, except during rush hours when the logging interval is 15 minutes. Hourly values were calculated for the whole measurement period. Global radiation and PAR from Kumpula are also included in Mäkelänkatu data set. Other aerosols, gaseous and meteorological data were also averaged hourly. For easier comparison, the data from all the instruments were synchronized to UTC+2.

Apart from measuring LDSA with the instruments (LDSA_ins_), LDSA parameter was calculated from DMPS (LDSA_cal_) by a conversion equation with alveolar deposition efficiency and summing up the total LDSA concentration of the aerosol. The particle size-dependent deposition fraction conversion equation was first introduced in the human respiratory tract model on the ICRP report by Bair in 1994 [48]. Kuula et al. [49] revealed that LDSA_cal_ is comparable to the instrumental measurements of LDSA_ins_ by Pegasor. Total particle number concentration (N_tot_) and particle modes were extracted from size distribution data. The four particle modes used in this study are nucleation mode, Aitken mode, adjusted Aitken mode, which is believed to be the range of BC size distribution [50], and accumulation mode. The range of the size bins is different in Mäkelänkatu and Kumpula, and the total particle number concentration of the whole spectrum of DMPS measurements represents particles with diameter between 6–800 nm and 3–950 nm, respectively. Nucleation mode sums up the particle number concentration of size bins below 0.025 µm (PN_0.025_). Aitken and adjusted Aitken modes were defined to include particles with diameter between 0.025–0.09 µm (PN_0.09-0.025_) and 0.030–0.30 µm (PN_0.3-0.03_), respectively. Accumulation mode was also calculated as size fractions 90 µm or above (PN_1-0.090_) in both sites. 

Since wind direction is a circular variable, we applied trigonometric function sine and cosine to resolve into North-South and East-West vector components, as demonstrated in other literature [19,24]. The numerical values wind direction variables, named as WD–N and WD–E respectively, are zero-direction and eligible for the application of standard linear methods [51]. Variables from two measurement sites were converted into the same unit. PM_2.5_, PM_10_, BC were calculated in mass concentration µg m^−^³. Particle modes were reported in number concentration cm^−3^ and LDSA in µm^2^ cm^−3^. All gaseous variables were measured in volume concentration ppb. 

In order to perform linear regression, response and predictor variables are assumed to be in normal distribution. By checking distribution histograms, most aerosols and gaseous variables, NO, NO_2_, NO_x_, BC, PM_10_, PM_2.5_, O_3_, SO_2_, CO, two LDSA variables, particle modes and traffic, have skewed distribution and logarithmic transformation were applied as performed in Fernández-Guisuraga’s paper in 2016 [24]. Unlike Equation (2), a log-link function was then used to specify the relationship between expected response and other predictors [24]:(9)$$\hat{Y}=exp\left(\beta_{0}+\sum_{i=1}^{p}\beta_{i}X_{i}\right)$$
where $\hat{Y}$ is the output variable, which is BC in this case, $\beta_{0}$ is the intercept of the model, and $\beta_{i}$ and $X_{i}$ correspond to the ith regression coefficient and ith input variable of the model, respectively.

### 3.4. Data Completeness

The two data sets have different levels of data completeness and their missing data distributions differ. Table 2 outlines the data completeness for the key parameters in the two data sets. In general, data in Mäkelänkatu are more complete (mostly above 90%), except for CO, LDSA_cal_ and particle modes. They cover only 45–80% in winter and spring. LDSA_cal_ and particle modes have the same data completeness because both are converted by DMPS mathematically. 

The data in Kumpula in the period of spring have relatively low completeness percentage, especially for NO_x_, BC, PM_10_ and LDSA (below 40%). The missing mechanism of most variables is MAR. Traffic data at Mäkelänrinne are used for both sites. The data completeness of traffic data ranges from 69% to 94% in different classes and the data gap is periodic. The meteorological data for both sites are sufficiently complete to support full imputation with the input-adaptive proxy method.

### 3.5. Classification

Since air pollutants show varying patterns due to their meteorological conditions and traffic patterns, we categorized the data into several classes based on seasons and workdays.

A definition of thermal (temperature-based) seasons was used. Spring and autumn are the periods when daily average temperatures are between 0 °C and 10 °C, and winter and summer are when the temperatures are below 0 °C and above 10 °C, respectively [19]. A sinusoid curve was fitted to the daily average temperature time series. The season was determined when the curve passed through the temperature threshold (Figure 3). According to this definition, winter was found to be from 1 January to 7 March 2017, from 10 January to 4 April 2018 and 13 December to 31 December 2018 with a total number of 171 days. Summer covered from 15 May to 28 September 2017 and from 10 May to 23 September 2018 with a total number of 274 days. The number of days in spring and autumn were 102 and 183, respectively.

Workdays and weekends are classified for each of the four seasons since they perform differently in terms of traffic rates and corresponding air pollutant concentrations [19,52]. Workday category typically includes all weekdays, excluding all public holidays, while weekend category represents all Saturdays and Sundays, plus public holidays. 502 workdays and 228 weekends were identified during the measurement period. Altogether, the classification generates eight classes and proxies were developed separately.

## 4. Results and Discussion

This section describes first the meteorological condition in year 2017 and 2018, in which our training data were measured. We also show the seasonal and diurnal pattern of BC and the correlation between BC and other variables in Section 4.1. Section 4.2 focuses on the evaluation of proxy performance.

### 4.1. Variable Characterization

Figure 3 presents daily averaged meteorological data for the whole measurement period. Since we used thermal definition to classify season cases, the length of seasons is highly dependent on the daily temperature. Compared with year 2017, the period of the seasonal cycle in 2018 is much shorter, which indicates that the seasonal changes came more quickly. The winter between the year 2017 and 2018 started late in early January in 2018 but the summer season came very early with a high amplitude. It shortened the spring season and lengthened the summer season in 2018. In addition, during 2018 there was a drought episode during May and June. The daily RH mean dropped below 50% for more than one month. Wind speed, air pressure and PAR are comparable for both years, ranging from 1 to 4 ms^−1^, from 975 to 1050 hPa and from 0 to 650 W m^−2^, respectively. The unexpected heat wave and drought in 2018 may affect the range of confident intervals when we fit the data to the models.

Figure 4 illustrates the monthly (y-axis) and diurnal (x-axis) patterns of measured BC in street canyon in Mäkelänkatu. BC concentrations behave slightly differently in all months on workdays and weekends so we will develop separate proxies. Figure 4a shows two peaks in BC concentration during workdays at 7–9 in the early morning and at 4–6 in the late afternoon in all months due to the elevated traffic counts during working peak hours [46,53,54]. The evening peak during workday is smaller than the morning peak because of the mixing height in the atmosphere [53,54]. The higher mixing height in the evening allows BC to mix into a larger air volume; therefore, a lower concentration at surface is measured. During weekends, only one peak is observed in the late evening (Figure 4b). The evening peak during weekends appears at 17–20 in the wintertime and at 20–22 in the summertime. The boosted nocturnal BC concentration might be owing to the increasing traffic rates along the daytime, reaching a peak approaching sunset when residents in the city return home. The peak of BC in weekends is much smaller than in workdays. The arithmetic mean of BC in weekends is also approximately 50% less than in workdays (Table 3). In terms of seasonality, considering only workdays, BC concentration in the summer (1.29 ± 0.22 µg m^−3^) has the highest value and spring BC (1.07 ± 0.52 µg m^−3^) has the lowest among all seasons. The difference between the highest and the lowest is about 20%, which is in alignment of the observation by Helin et al. [35] who suggested the lack of BC seasonal variation in traffic environment.

As an urban background station in Kumpula, the source of BC is more varied, not only vehicle exhaust from traffic roads, but also residential wood combustion. Since there is a forest between the station and a nearby road, it may block part of the BC from traffic. The long distance from the source could also increase dilution rate of BC. The two possible factors enhance the relative influence from residential activity. The discrepancies between workdays and weekends are smaller (10–20%) in all seasons because the direct effects by traffic is milder than Mäkelänkatu. Winter appears to have the highest BC (0.67 ± 0.38 µg m^−3^) among all seasons because of the lower mixing and elevated wood combustion by domestic heating [55] (Table 3). Another possible reason is that cold-start of a car engine at low ambient temperature may enhance the BC emissions from light-duty vehicles [56]. The overall BC mass concentration is lower than in street canyon by 30% in the winter and more than 50% in the summer during workdays.

In order to find out the most linearly correlated terms with BC, we computed their absolute Pearson correlation coefficient as in two matrixplots (Figure 5). The redness of each box represents the degree of correlation. The logarithm of BC is highly correlated with part of the gaseous and aerosol variables in both sites. 

The correlations are fairly good in all eight classes in street canyon, in particular, the three NO_x_ variables, particle modes and the two LDSA variables (R = 0.5–0.9). We also notice that the correlation on workdays is much higher than that on weekends. It is due to the fact that a great portion of NO_x_ and the number of particles also come from traffic as BC emissions do [35]. Therefore, the amount of traffic is supposed to be able to explain for the discrepancy between the case on workdays and weekends. Unexpectedly, the correlations with traffic rates are just moderate (R = 0.3–0.7). The reason might be that the traffic counter is few hundred meters north to the supersite and the count does not fully represent the number of vehicles passing the station. Additionally, the traffic counter does not separate between different types of vehicles. Another reason for the low impact of the traffic counts might be that the weather conditions cause fluctuation. Even so, the difference in correlation between workdays and weekends are more significant than other variables in all seasons. 

In urban background, the correlations of BC with gaseous compounds and aerosols are generally lower. No clear discrepancies between workdays and weekends are observed. However, in winter and spring, the correlations of BC with the two LDSA variables and accumulation mode are unexpectedly high. The adjusted Aitken mode, unexpectedly, does not exhibit a noticeable correlation with BC as suggested in previous study [50]. The reason might be that the correlation relationship is location-specific and the source of BC in different cities might not be the same.

The linear association of BC and meteorological data remain low in both sites [57]. Wind speed ranks top (R = 0.2–0.5) among the other meteorological variables [34,54]. On the other hand, the two wind direction variables (WD–N and WD–E) perform very little correlation with BC (R < 0.1). In street canyon, wind speed appears to correlate better with BC in weekends (R = 0.3–0.5) than in workdays (R = 0.2–0.3) while radiation variables perform in an opposite way. No clear discrepancies between workdays and weekends are observed in the urban background environment. 

Figure 6 presents the combined R scores without any classification for every measured variable in descending order in both sites. Irrelevant variables (R < 0.1) were discarded from the whole data set and were omitted from further analysis in order to minimize the computational time. We did not pick a higher threshold because the variables with low R scores might act as an accessory input variable in proxy models in the combination with other more-correlated variables. Eventually, we selected 19 variables in both Mäkelänkatu and Kumpula for the model creation step.

### 4.2. Performance of the Imputation Proxies

In general, the traditional proxy in the street canyon gives fairly good continuous estimations, with $adjR^{2}$ ranging from 0.86 to 0.94 (Table 4), MAE mostly less than 0.01 and RMSE ranging from 0.19 to 0.28. The value of $adjR^{2}$ is slightly higher in workdays than in weekends. The reason might be that the number of workday samples is much higher than that weekend samples, which lowers the uncertainties. No obvious seasonal discrepancies are found in terms of model evaluation. A noticeably high MAE, however, is found in autumn weekends. After ranking the different models by evaluation attributes described in Section 2.3, the best performing OLS models give the same results in seven out of eight classes, such that all three input variables are traffic counts, NO_x_ and accumulation mode. The high resemblance is because BC source in street canyon is mainly fuel combustion from traffic, which at the same time emits NO_x_ to the atmosphere, despite the time variation. The only exception is found in winter weekends, in which CO appears to better explain for BC instead of NO_x_. Due to the similarities in the model selection in all classes, classification may not be needed in Mäkelänkatu to maintain its simplicity.

In Kumpula, the urban background station gives a slightly worse $adjR^{2}$ than in the street canyon, ranging from 0.74 to 0.91 (Table 5). The MAE in spring and autumn weekends is exceptionally high, 0.056 and 0.036, respectively. The RMSE in spring and summer is higher than 0.4, which indicates more largely biased fitted points are found in these seasons. Unlike street canyon, the input variables that explain BC are varying in urban background station. Accumulation mode appears to play an important role in explaining BC in most classes. CO and LDSA are also significant in certain seasons. Temperature is the only meteorological input variables to be included in the best performing regression models. Due to the rule of only including three input variables to the model and various BC sources in urban background, the regression proxy could not include all relevant input variables to the models. Besides, in this data set, we did not include any variables as a direct indicator to residential activity; therefore, the performance is poorer. 

Note that the above only show the performance of the traditional proxy, i.e., the best performing model in the input-adaptive proxy. Figure 7 better illustrates how different the two proxies behave. During the period from 1 January to 20 January 2018, the proxy BC fits fairly well and follows the daily cycle. However, it occasionally underestimates at peaks and overestimates at toughs compared to the original data (black line). This serves as one of the drawbacks of using robust OLS fitting in linear regression because this technique penalizes outliers and makes them less influential to the regression in order to optimize the estimation of regression slopes [28]. In this way, the robust method sacrifices the estimation at extreme data points. Additionally, the estimation by traditional proxy is not always available and, in this study, only 64.5% of BC concentration in Mäkelänkatu (Figure 7a) and 54.7% in Kumpula (Figure 7b) are estimable by the method. 

The missing gap will then be substituted by input-adaptive proxy (red line), such that proxy BC gives continuous estimation. The fitting performance does not show obvious difference in the classification of workdays and weekends. The performance in the two locations does not differ either. On top of the diurnal pattern on individual days, both traditional and input-adaptive proxies estimate the averaged diurnal pattern with a satisfactorily high accuracy (Figure 8). 

In Mäkelänkatu, the diurnal fitting by both traditional and input-adaptive proxies for workdays in all seasons performs well (Figure 8a). The highest deviation in the diurnal cycle takes place in winter weekends (Figure 8b) where only about 20% of data is fitted with the traditional model. However, the input-adaptive proxy shows very similar diurnal cycle as the measured BC in the same class. This is because all data are fitted with the top 10 best models in the input-adaptive proxy (Figure 9), so $adjR^{2}$ remains high. 

In Kumpula, the estimation has a greater off in diurnal cycle for both proxies with bigger standard derivation. Figure 8c shows a bigger deviation in urban background between the original BC diurnal pattern and that by both proxies compared to street canyon environment. The input-adaptive proxy also appears to have a shift in estimating morning peak in spring and summer workday. The peaks by the input-adaptive proxy appear 1–2 hours earlier than the original data. The reasons remain unknown. For weekends, although there are no as clear diurnal patter as in the others, both traditional and input-adaptive proxies do not model the original data as well as the others, except the winter classification (Figure 8d). Overestimation of traditional proxy BC is found in spring and autumn weekends, and it might be attributed to the insufficiency of successful estimation in each hour and thus influencing the averaged values. Only less than 20% of data are successfully estimated in these two classes (Figure 9). Figure 9 shows the probability of data points fitted by models of input-adaptive proxy with different ranks in both locations. In Mäkelänkatu, about 20–90% of data points are fitted by the best model depending on different seasons (Figure 9a). The pattern of model ranking probability is similar on both workdays and weekends in all seasons except winter. While the best model can estimate 65% of the total data points in winter workdays, it is only able to fit less than 20% in winter weekends. However, the rest of the data points in winter weekends manage to be estimated by the other top ten best models; therefore, the overall coefficient of determination remains high. In the urban background site in Kumpula, the probability of using the best model is much lower on weekends than on workdays in all seasons (Figure 9b). Winter appears to have the highest percentage of estimation by the best model. The other seasons do not show obvious deviation.

Table 6 shows the input variables and the corresponding evaluation attributes in winter workdays in Mäkelänkatu as an example. Accumulation mode, which was found to resemble the pattern of BC in Helsinki in previous studies [34,58], is found in every single model in the table. Other variables are NO_x_, NO_2_, traffic counts and some meteorological variables. Although meteorological variables have relatively low correlation coefficient R with BC, they still manage to play a role as an accessory input variable in the top ten models. Because of the lower dependency of meteorological data with other air quality variables [57], the meteorological variables survive in the model rejection tests and guarantee the estimation in input-adaptive proxy. This serves an example to suggest that individual R does not necessarily reflect the contribution in models with more than one input variable.

Furthermore, the evaluation attributions in the top ten models are very close to each other. The $adjR^{2}$ scores in the top ten models only vary in a range of 2%. If we choose MAE or RMSE over $adjR^{2}$ to be the first ranking criteria, the outcomes of the model rank become very different.

The overall performance difference in the street canyon in Mäkelänkatu and the urban background in Kumpula indicates that input-adaptive proxy is location-specific because of their different pollutant sources. Street canyon has been suggested to have relatively constant BC source [35], so the regression performance in street canyon is overall better than in urban background. Model performance also depends on the characteristics of missing data patterns, like gap lengths and number of complete rows, but not the amount of missing data, suggested by Junninen et al. [4] and Junger and Ponce de Leon [9].

However, the proxy accuracy might drift when we train the same type of data set in different periods. Therefore, we should train longer historical data to lower the uncertainties of the proxy. The prolonged drought in year 2018 might influence the BC variation and thus reducing the proxy reliability in the future. 

Since this input-adaptive is simple yet reasonable and manages to give continuous estimation, it is applicable to fill up gaps in air quality data and act as a virtual sensor of air pollutant. In case one of the input variables have measurement failure, we can still model the output variable with a relatively reliable estimate. Ideally, this input-adaptive proxy could co-exist side-by-side with on-site measurements in support of each other. In case of unavailability of on-site measurements, input-adaptive proxy can act as a virtual sensor. The computed output variable, in this case BC, is also useful in filling in missing pieces in air quality index.

## 5. Conclusions

In this study, we presented a modified method to impute missing data by input-adaptive proxy, which manages to fill up all the voids and give reasonable estimates. We used BC concentration in Mäkelänkatu traffic site and Kumpula urban background site as a case study. We then created models with selected input variables, which were determined by Pearson correlation coefficient. We evaluated and ranked the models by their $adjR^{2}$, MAE and RMSE. By using the traditional proxy method, about 20–80% of missing BC data was filled up. The input-adaptive proxy selected input variables and managed to fill up all the missing voids. In addition, the overall performance is reliable, with $adjR^{2}$ of 0.86 to 0.94 in the street canyon and 0.74 to 0.91 in the urban background. Its flexibility and reliability ascertain for the input-adaptive proxy to act as a virtual sensor.

Regression performance is better in the street canyon because BC can be mostly explained by traffic counts, NO_x_ and accumulation mode. The BC source in urban background is heterogeneous, which includes traffic and residential wood burning. It is also affected by some meteorological parameters. The diurnal pattern in BC is clearly different on workdays from weekends. The proxy performance in workdays is slightly better than in weekends in both sites. In the street canyon, the proxy works quite similarly in all seasons, while in the urban background station, the proxy performance in spring and summer is poorer than the other seasons. Although the overall regression performance seems promising, the results can be altered by choosing other ranking criteria in the model evaluation scheme.

The input-adaptive proxy could be modified by inserting autoregression term [15,23,59] or time variables [4,10] because the dataset is a time series and air quality variables have a strong time dependency. Although this modification would add slight computational complexity to the model, the eight classes in this study are no longer needed. Another modification is that we could investigate their mutual information for non-linear relationship developed by Zaidan et al. [60], instead of checking the linear association among the air quality variables.

## Figures and Tables

**Figure 1 sensors-20-00182-f001:**
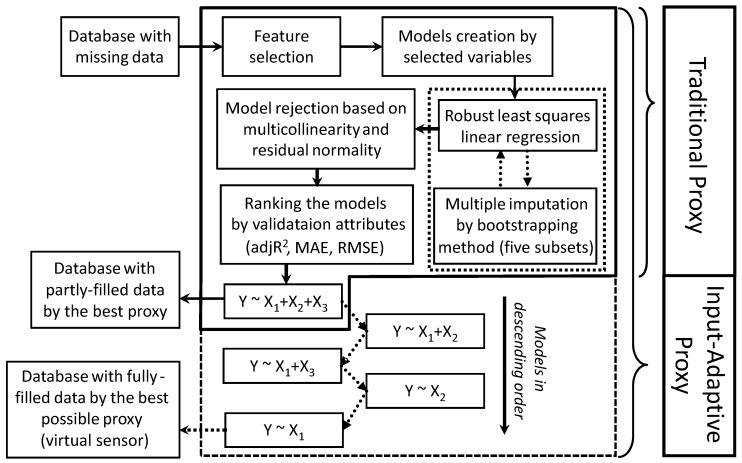
Schematic diagram of the procedures of developing the recursive input-adaptive proxy.

**Figure 2 sensors-20-00182-f002:**
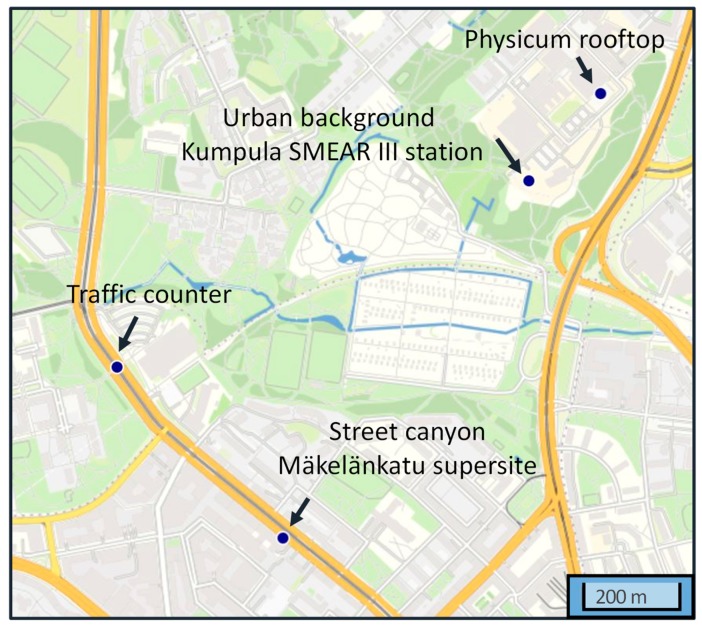
The location of the measurement sites in Helsinki, Finland.

**Figure 3 sensors-20-00182-f003:**
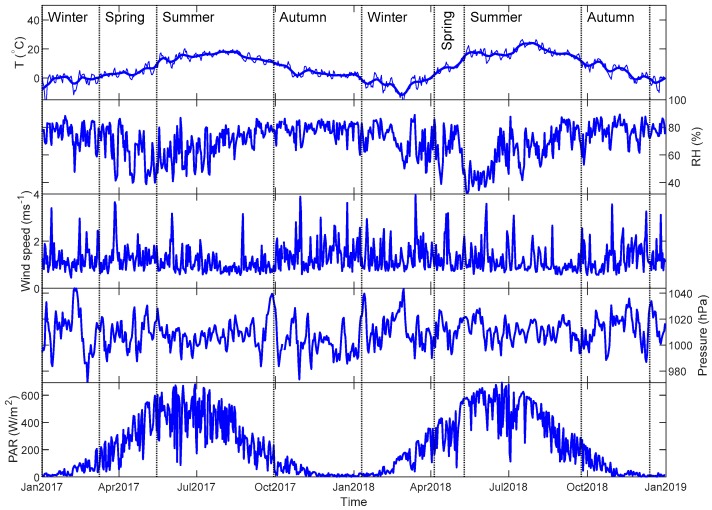
Meteorological data during the measurement period. Daily temperature (°C), relative humidity (%), wind speed (m s^−1^), pressure (hPa) and PAR (W m^−2^) are shown from the top panel to the bottom. A smoothing curve with lowess filter is also plotted on top of the daily temperature to identify the seasons based on thermal definition. Vertical lines clarify the separation of seasons.

**Figure 4 sensors-20-00182-f004:**
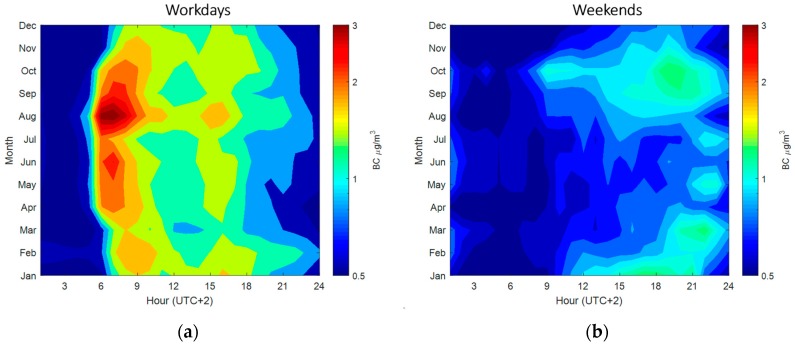
Contour plot of diurnal BC concentration (µg m^−3^) for each month in Mäkelänkatu street canyon. X-axis denotes the hour of day in UTC+2 and y-axis denotes months from January to December. (**a**) illustrates the variation in workdays and (**b**) represents weekends. Color bar is in logarithmic scale, where warm colors indicate higher concentrations and cold colors indicate lower concentrations.

**Figure 5 sensors-20-00182-f005:**
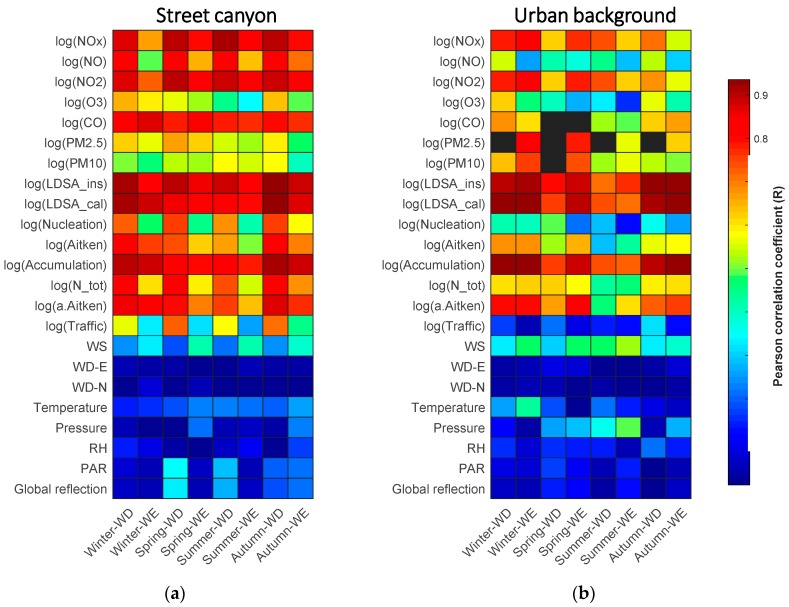
Matrixplots showing the Pearson correlation coefficients of the logarithm of BC with other variables in eight classes as shown in the columns. (**a**,**b**) illustrate the data in the street canyon in Mäkelänkatu and the urban background in Kumpula, respectively. The redness of grid represent the degree of correlation. Blue colors show the least correlated variable combination. Black color represents the data in that period is missing.

**Figure 6 sensors-20-00182-f006:**
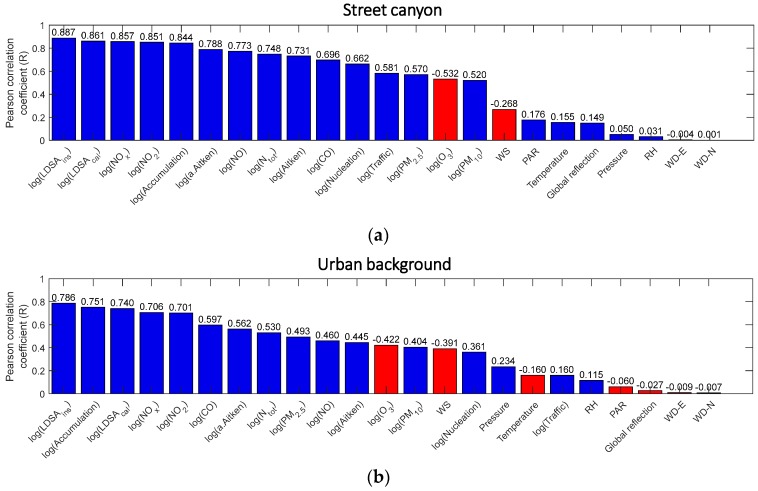
Bar charts showing the overall Pearson coefficient coefficients of logarithm of BC with other variables without any classification. The variables are arranged in descending order of the absolute value of R. Numeric R scores are presented on top of each bar. Negative R scores are illustrated by the red color bars. (**a**,**b**) show the data in street canyon in Mäkelänkatu and in urban background in Kumpula, respectively.

**Figure 7 sensors-20-00182-f007:**
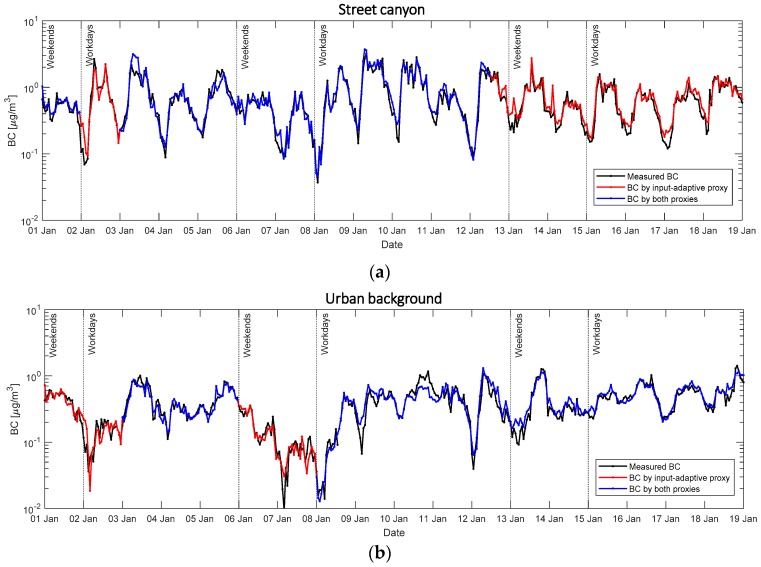
Time series plots in the period from 1 January to 20 January 2018 showing measured BC concentrations in black line. Blue lines represents the calculated BC by the traditional proxy (the input-adaptive proxy with the best performance). Red lines shows the calculated BC by input-adaptive proxy when the estimation could not be made by the traditional proxy. (**a**,**b**) show data in Mäkelänkatu and Kumpula respectively. Y-axis shows BC concentration in µg m^−^³ shown in logarithmic scale. Workdays and weekends separation are indicated by vertical lines.

**Figure 8 sensors-20-00182-f008:**
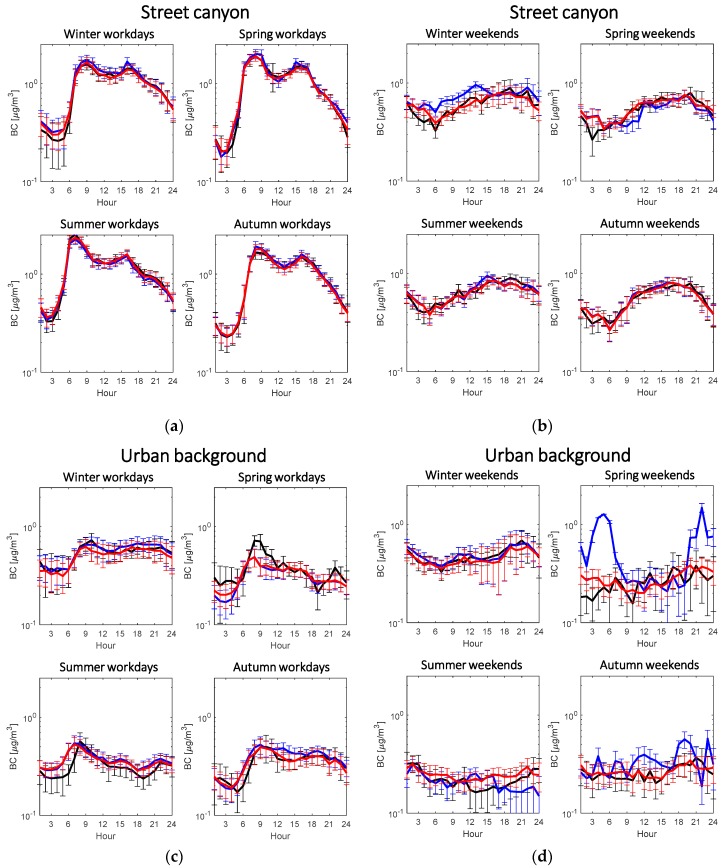
Four diurnal subplots of original measured BC (black line), BC calculated by traditional proxy (blue line) and BC calculated by input-adaptive proxy (red line). (**a**,**b**) show diurnal pattern in the street canyon in Mäkelänkatu on workdays and on weekends, respectively. (**c**,**d**) show diurnal pattern in the urban background station in Kumpula on workdays and on weekends, respectively. Error bars are shown as standard deviation at each hour of the day.

**Figure 9 sensors-20-00182-f009:**
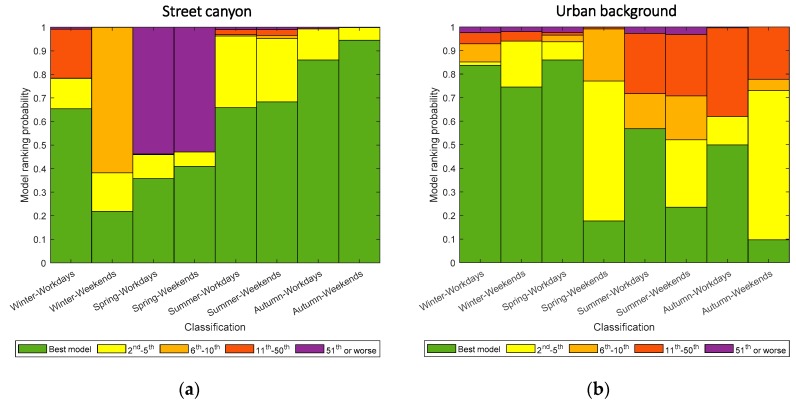
The probability of data points fitted by models of different rank categories (as in legend) used in input-adaptive proxy for different time classes. (**a**,**b**) represent the probability in Mäkelänkatu and Kumpula, respectively.

**Table 1 sensors-20-00182-t001:** Summary of the measured parameters and their instruments used in Mäkelänkatu and Kumpula. Same instrument was used in both locations for BC and LDSA measurement.

Measured Quantity	Instruments in Mäkelänkatu	Instruments in Kumpula
Aerosols		
BC	MAAP Thermo Scientific 5012 with a PM_1_ inlet	
LDSA	Pegasor AQ Urban sensor	
PM_2.5_ and PM_10_	Continuous ambient particulate monitor Thermo TEOM 1405	Continuous ambient particulate monitor Thermo TEOM 1405-D
Particle size distribution	Single DMPS (Vienna DMA and Airmodus A20 CPC)	Twin DMPS (Hauke-type DMA and TSI Model 3025 CPC + Hauke-type DMA and TSI Model 3010 CPC)
Trace gases		
NO, NO_2_ and NO_x_	Chemiluminescence analyzer Horiba APNA-370	Chemiluminescence analyzer Thermo TEI42S
O_3_	UV photometric analyzers Horiba APOA-370 and Thermo Model 49i	IR-absorption photometer TEI49
CO	IR-absorption analyzer Horiba APMA-360	IR-absorption analyzer Horiba APMA-370
SO_2_	N/A	UV-fluorescence analyzer Horiba APSA-360
Meteorological variables		
Wind speed and wind direction	Weather transmitter Vaisala WXT 520 and Vaisala WXT536	Vaisala cup anemometer
Air temperature		Platinum resistant thermometer Pt-100
RH		Platimun resistance thermometer and thin film polymer sensor Vaisala DPA500
Air pressure		Barometer Vaisala DPA500
Global radiation and PAR	N/A	Kipp and Zonen CNR1+PAR lite

**Table 2 sensors-20-00182-t002:** Data completeness (%) of the two data sets pertaining to the key air quality and meteorological parameters in Mäkelänkatu and Kumpula (in parenthesis). The last row shows number of days that fall on the category. WD and WE are the abbreviations of workdays and weekends classes.

	Winter		Spring		Summer		Autumn	
	WD	WE	WD	WE	WD	WE	WD	WE
BC	99 (59)	100 (64)	100 (35)	100 (32)	99 (75)	99 (78)	99 (89)	100 (89)
O_3_	98 (99)	100 (100)	98 (99)	99 (100)	98 (100)	99 (98)	99 (100)	100 (100)
CO	59 (87)	64 (85)	69 (65)	76 (68)	96 (72)	95 (71)	95 (100)	96 (100)
NO_x_	99 (98)	100 (99)	99 (34)	99 (32)	99 (83)	99 (80)	99 (100)	100 (99)
PM_10_	98 (47)	99 (50)	98 (35)	98 (32)	98 (48)	98 (49)	99 (43)	98 (39)
LDSA_ins_	98 (51)	99 (47)	95 (35)	91 (32)	99 (52)	97 (51)	95 (55)	99 (60)
LDSA_cal_	79 (88)	75 (79)	46 (71)	47 (78)	98 (76)	97 (64)	100 (56)	100 (65)
Mode	79 (98)	75 (89)	46 (97)	47 (100)	98 (99)	97 (95)	100 (98)	100 (100)
Temperature	100 (100)	100 (100)	100 (100)	100 (100)	99 (100)	99 (100)	99 (100)	100 (100)
Traffic ^1^	86	82	88	94	69	72	87	94
n	116	55	68	32	192	82	126	57

^1^ Traffic data were collected in proximity to Mäkelänrinne swimming centre, about 600 m south and 1 km north from Kumpula and Mäkelänkatu measurement sites, respectively.

**Table 3 sensors-20-00182-t003:** Mean, standard deviation (SD) and 10th, 25th, 50th, 75th and 90th percentiles (P10, P25, P50, P75 and P90) of BC concentration (µg m^−3^) in four seasons on both workdays and weekends in the street canyon in Mäkelänkatu, denoted by WD and WE, respectively. Data of urban background environment in Kumpula are presented in parenthesis.

	Winter		Spring		Summer		Autumn	
	WD	WE	WD	WE	WD	WE	WD	WE
Mean	1.10 (0.67)	0.76 (0.60)	1.07 (0.52)	0.66 (0.47)	1.29 (0.42)	0.75 (0.33)	1.12 (0.47)	0.69 (0.37)
SD	0.32 (0.38)	0.53 (0.66)	0.57 (0.79)	0.64 (2.10)	0.22 (0.11)	0.27 (0.22)	0.30 (0.17)	0.40 (0.32)
P10	0.22 (0.17)	0.26 (0.14)	0.20 (0.13)	0.20 (0.08)	0.31 (0.10)	0.22 (0.07)	0.21 (0.10)	0.18 (0.06)
P25	0.43 (0.29)	0.40 (0.24)	0.41 (0.22)	0.31 (0.12)	0.56 (0.17)	0.39 (0.12)	0.44 (0.19)	0.32 (0.12)
P50	0.87 (0.53)	0.62 (0.48)	0.88 (0.37)	0.54 (0.27)	1.04 (0.31)	0.64 (0.21)	0.90 (0.35)	0.54 (0.27)
P75	1.51 (0.85)	0.92 (0.77)	1.50 (0.66)	0.82 (0.61)	1.74 (0.54)	0.99 (0.42)	1.54 (0.60)	0.884 (0.45)
P90	2.25 (1.20)	1.29 (1.13)	2.14 (1.16)	1.25 (1.05)	2.61 (0.87)	1.38 (0.74)	2.31 (0.98)	1.42 (0.81)

**Table 4 sensors-20-00182-t004:** Best performing OLS regression traditional proxy results in the street canyon in Mäkelänkatu (all models and all coefficients of the corresponding input variables have *p* < 0.05). X represents the predictor variables in the different classes. β and SE are the coefficients of the estimates and the corresponding standard errors, respectively.

	X	β	SE	adjR^2^	MAE	RMSE
Winter Workdays	log(Traffic)	0.07	0.01	0.92	0.008	0.26
	log(NO_x_)	0.37	0.01			
	log(PN_1-0.09_)	0.67	0.01			
Winter Weekends	log(Traffic)	0.08	0.02	0.89	0.003	0.19
	log(CO)	1.00	0.11			
	log(PN_1-0.09_)	0.63	0.04			
Spring Workdays	log(Traffic)	0.13	0.02	0.93	0.006	0.24
	log(NO_x_)	0.44	0.02			
	log(PN_1-0.09_)	0.64	0.03			
Spring Weekends	log(Traffic)	0.16	0.02	0.87	0.007	0.23
	log(NO_x_)	0.42	0.02			
	log(PN_1-0.09_)	0.54	0.02			
Summer Workdays	log(Traffic)	0.12	0.01	0.92	0.008	0.24
	log(NO_x_)	0.55	0.01			
	log(PN_1-0.09_)	0.48	0.01			
Summer Weekends	log(Traffic)	0.11	0.01	0.86	0.004	0.28
	log(NO_x_)	0.49	0.01			
	log(PN_1-0.09_)	0.60	0.02			
Autumn Workdays	log(Traffic)	0.07	0.01	0.94	0.004	0.23
	log(NO_x_)	0.38	0.01			
	log(PN_1-0.09_)	0.67	0.01			
Autumn Weekends	log(Traffic)	0.10	0.01	0.91	0.016	0.25
	log(NO_x_)	0.37	0.01			
	log(PN_1-0.09_)	0.74	0.01			

**Table 5 sensors-20-00182-t005:** Best performing OLS regression traditional proxy results in the street urban background in Kumpula (all models and all coefficients of the corresponding input variables have *p* < 0.05). X represents the predictor variables in the different classes. β and SE are the coefficients of the estimates and the corresponding standard errors, respectively.

	X	β	SE	adjR^2^	MAE	RMSE
Winter Workdays	log(NO_2_)	0.19	0.01	0.91	0.008	0.23
	log(CO)	0.54	0.05			
	log(PN_1-0.09_)	0.77	0.02			
Winter Weekends	log(CO)	0.23	0.11	0.88	0.006	0.27
	log(PN_1-0.09_)	1.00	0.02			
	log(PN_0.025_)	0.16	0.01			
Spring Workdays	log(Traffic)	0.06	0.03	0.74	0.013	0.46
	log(PN_1-0.09_)	0.95	0.04			
	log(PN_0.025_)	0.28	0.03			
Spring Weekends	Temperature	−0.05	0.01	0.84	0.056	0.43
	log(NO)	−0.06	0.03			
	log(LDSA_cal_)	1.23	0.05			
Summer Workdays	log(NO_x_)	0.33	0.01	0.77	0.007	0.40
	log(CO)	0.77	0.07			
	log(PN_1-0.09_)	0.58	0.02			
Summer Weekends	log(CO)	1.41	0.12	0.78	0.001	0.43
	log(PM_10_)	0.05	0.03			
	log(PN_1-0.09_)	0.92	0.04			
Autumn Workdays	log(Traffic)	0.14	0.01	0.91	0.014	0.25
	log(PN_0.09-0.025_)	−0.46	0.02			
	log(LDSA_ins_)	1.46	0.02			
Autumn Weekends	Temperature	−0.04	0.08	0.90	0.036	0.27
	log(PM_10_)	−0.01	0.01			
	log(LDSA_cal_)	1.34	0.04			

**Table 6 sensors-20-00182-t006:** Top ten best models in street canyon Mäkelänkatu winter workdays.

	X_1_	X_2_	X_3_	adjR^2^	MAE	RMSE
1	log(PN_1-0.09_)	log(Traffic)	log(NO_x_)	0.92	0.008	0.2602
2	log(PN_1-0.09_)	log(NO_x_)	Wind speed	0.92	0.010	0.2600
3	log(PN_1-0.09_)	log(NO_x_)	Air pressure	0.92	0.011	0.2568
4	log(PN_1-0.09_)	log(NO_x_)	PAR	0.92	0.013	0.2594
5	log(PN_1-0.09_)	log(NO_x_)	Temperature	0.92	0.014	0.2558
6	log(PN_1-0.09_)	log(Traffic)	log(CO)	0.92	0.017	0.2545
7	log(PN_1-0.09_)	log(Traffic)	log(NO_2_)	0.92	0.018	0.2567
8	log(PN_1-0.09_)	log(NO_2_)	Wind speed	0.91	0.001	0.2454
9	log(PN_1-0.09_)	log(NO_x_)	log(PM_2.5_)	0.91	0.002	0.2535
10	log(PN_1-0.09_)	log(NO_2_)	Air Pressure	0.91	0.006	0.2495

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
