# Peer review of "Input-Adaptive Proxy for Black Carbon as a Virtual Sensor"

_sensors, 2019, doi:10.3390/s20010182_

Round 1

Reviewer 1 Report

The research design seems to be appropriated and the adopted methods are well and extensively described. The results are clearly presented and they support the conclusions; the next required development of the methodology are presented, too.

In this study, the authors present a modified method to impute missing data by input-adaptive proxy, which manages to fill up all the voids and give reasonable estimates. They used BC concentration in Mäkelänkatu traffic site and Kumpula urban background site as a case study. They then created models  with selected input variables, which were determined by Pearson correlation coefficient. They evaluated and ranked the models by their ????2, ??? and ????. By using the traditional proxy  method, about 20–80% of missing BC data was filled up. The input-adaptive proxy selected input variables and managed to fill up all the missing voids. In addition, the overall performance is reliable,  with ????2 of 0.86 to 0.94 in the street canyon and 0.74 to 0.91 in the urban background. Its flexibility and reliability ascertain for the input-adaptive proxy to act as a virtual sensor. 
Regression performance is better in the street canyon because BC can be mostly explained by  traffic count, NOx and accumulation mode. The BC source in urban background is heterogeneous, which includes traffic and residential wood burning. It is also affected by some meteorological  parameters. The diurnal pattern in BC is clearly different on workdays from weekends. The proxy  performance in workdays is slightly better than in weekends in both sites. In the street canyon, the  proxy works quite similarly in all seasons, while in the urban background station, the proxy  performance in spring and summer is poorer than the other seasons. Although the overall regression performance seems promising, the results can be altered by choosing other ranking criteria in the  model evaluation scheme.

The paper, from my point of view, can be published.

Author Response

Response to Comments by Reviewer #1

Comments and Suggestions for Authors

The research design seems to be appropriated and the adopted methods are well and extensively described. The results are clearly presented and they support the conclusions; the next required development of the methodology are presented, too.

In this study, the authors present a modified method to impute missing data by input-adaptive proxy, which manages to fill up all the voids and give reasonable estimates. They used BC concentration in Mäkelänkatu traffic site and Kumpula urban background site as a case study. They then created models  with selected input variables, which were determined by Pearson correlation coefficient. They evaluated and ranked the models by their ????2, ??? and ????. By using the traditional proxy  method, about 20–80% of missing BC data was filled up. The input-adaptive proxy selected input variables and managed to fill up all the missing voids. In addition, the overall performance is reliable,  with ????2 of 0.86 to 0.94 in the street canyon and 0.74 to 0.91 in the urban background. Its flexibility and reliability ascertain for the input-adaptive proxy to act as a virtual sensor.

Regression performance is better in the street canyon because BC can be mostly explained by  traffic count, NOx and accumulation mode. The BC source in urban background is heterogeneous, which includes traffic and residential wood burning. It is also affected by some meteorological  parameters. The diurnal pattern in BC is clearly different on workdays from weekends. The proxy  performance in workdays is slightly better than in weekends in both sites. In the street canyon, the  proxy works quite similarly in all seasons, while in the urban background station, the proxy  performance in spring and summer is poorer than the other seasons. Although the overall regression performance seems promising, the results can be altered by choosing other ranking criteria in the  model evaluation scheme.

The paper, from my point of view, can be published.

Response: We thank the reviewer for pointing out the scientific value of this research paper and recommending publishing in “Sensors”. We made minor revisions according to the reviewers’ reports.

Reviewer 2 Report

In this manuscript, the authors develop an input-adaptive proxy, which selects input variables of other air quality variables based on their correlation coefficients with the output variable. Using the input-adaptive proxy, in this work the authors estimated black carbon (BC) concentration in two sites in Helsinki, as a case study, which respectively represent street canyon and urban background scenario.

It is a scientifically interesting and well-grounded work with a relevant practical application.

I have some concerns about the work presented which are more lines of future research than corrections to the manuscript:
1. A more careful identification of emitting sources should have been made, in particular in situations where the most powerful emitting source (the "main road") decreases in relative importance giving room for other sources to gain importance. Since these smaller sources are not clearly identified and described (in time and location), it has been difficult to explain the observed results, as well as to construct a more "tailored" model for end-of-week situations, for example;
2. I am not sure if applying the logarithm on the variables was sufficient to ensure their normalization. Was normalization verified prior to the decision to opt for parametric rather than nonparametric tests?
3. Given the available meteorological information, it would have been interesting to have a more robust description of dispersion conditions by estimating atmospheric stability conditions.

Please revise some minor edition details.

Author Response

Response to Comments by Reviewer #2

In this manuscript, the authors develop an input-adaptive proxy, which selects input variables of other air quality variables based on their correlation coefficients with the output variable. Using the input-adaptive proxy, in this work the authors estimated black carbon (BC) concentration in two sites in Helsinki, as a case study, which respectively represent street canyon and urban background scenario.

It is a scientifically interesting and well-grounded work with a relevant practical application.

Response: We thank the reviewer for pointing out the scientific value of this manuscript.

I have some concerns about the work presented which are more lines of future research than corrections to the manuscript:

A more careful identification of emitting sources should have been made, in particular in situations where the most powerful emitting source (the "main road") decreases in relative importance giving room for other sources to gain importance. Since these smaller sources are not clearly identified and described (in time and location), it has been difficult to explain the observed results, as well as to construct a more "tailored" model for end-of-week situations, for example;

Response: The emitting source at street canyon was mainly from traffic. Therefore our approach was to describe the variation in the amount of traffic and also meteorological conditions in different time. At the urban background site, as you said, the possible source is more varied, including traffic, domestic wood combustion, etc. The importance of source depends on the wind direction and also mixing layer. The models are separated into workdays and weekends mainly because of the difference in traffic counts. The prospective reviewer is right that a more tailor-made models can be constructed if we could also identify the minor source. It might be enough for the first paper of this method. In our future research, more different situations could be considered. Thank you for your suggestion.

I am not sure if applying the logarithm on the variables was sufficient to ensure their normalization. Was normalization verified prior to the decision to opt for parametric rather than nonparametric tests?

Response: We checked normalization by manually revising their distribution histogram before the decision. They are seemingly in normal distribution after the logarithm transformation. we also tested other transformations. It seems that logarithm is the best for this purpose. Furthermore, the fitting is done in a robust way to minimize the effect it brings if the data distribution is not normal enough. In the future, a proper test should be introduced in an automatic way. We have now added explanation for the choice of logarithmic transformation in the paper.

Given the available meteorological information, it would have been interesting to have a more robust description of dispersion conditions by estimating atmospheric stability conditions.

Response: Since we also mentioned mixing height and dispersion, it would be also helpful to include stability parameter z/L, or equivalent. However, this paper focuses more on methodology, so we did not describe so much on the atmospheric principles here. For the next step, when we write a paper on how meteorological conditions affect the performance of input-adaptive proxy, atmospheric stability conditions can be included as one of the input variables. Thanks for your suggestion.

Please revise some minor edition details.

Response: Together with the suggestions by other reviewers, we made minor revisions accordingly.

Reviewer 3 Report

This work presents an adaptive proxy procedure applied to black carbon, BC, measurements that can be used to fill up data gaps in measurements and to complement on-site observations.

The technique uses regressions methods as basis to propose modified methods, and they are both evaluated against BC datasets.

The paper is well structured. However, I advise to explain, reference, clearly specify and further back up the statistical terms, concepts and models used.  In particular:

the input-adaptive proxy model procedure should be more explicitly described. You should better support your assumption of having skew distributions for aerosol and gaseous parameters, and motivate your choice of using logarithmic transformations in details.

Some examples follow below.

Ln 47: “Currently, 92% of the world’s population lives in areas where air pollution concentration exceeds safe limits.” WHO Guidelines for healthier air? Please quote.

Ln 56: selection bias, please explain

Ln 59: “imputation by median”: please explain imputation

Ln 92: “Furthermore, ..virtual sensor”: Why? Please explain this clearly

Ln 82-83: “the deployment of this concept.. instruments”: this sentence needs to be backed up by more evidences.

Ln 84—88: “the purpose…sensors”. You need to be more specific.

Figure 1: The procedures of developing the recursive input-adaptive proxy cannot by fully understood from the picture.

Ln 117: ref?

Ln 118: “each model generated” which model? Please explain.

Ln 124: “bisquare weight function”. Please make more explicit.

Ln 126: “robust fitting”. Please describe in more details.

Ln 177: “valid data point”. Please explain.

Ln 304-305: “Since.. respectively”. Why? Please provide more details.

Section 3.2 Instruments:

You could add a table for the parameters measured and their instruments.

Ln 290: “conversion equation” please specify.

Ln 304-305: Please motivate this choice better. Why did you do this?

Ln 311: “Skew distributions”: Please explain this in more details and support with references.

Ln 336: “Thermal”, Temperature-based?

Author Response

Response to Comments by Reviewer #3

Comments and Suggestions for Authors

This work presents an adaptive proxy procedure applied to black carbon, BC, measurements that can be used to fill up data gaps in measurements and to complement on-site observations.

The technique uses regressions methods as basis to propose modified methods, and they are both evaluated against BC datasets.

Response: Many thanks for pointing out the scientific value of this work.

The paper is well structured. However, I advise to explain, reference, clearly specify and further back up the statistical terms, concepts and models used.  In particular:

the input-adaptive proxy model procedure should be more explicitly described. You should better support your assumption of having skew distributions for aerosol and gaseous parameters, and motivate your choice of using logarithmic transformations in details.

Response: Thank you for the comments and suggestions. The procedure of the model has more elaboration and the systematic diagram has also been modified for easier understanding. The skewed distribution for aerosol and gas parameters have now been supported by two references. we also included the step we made manually in the same paragraph.

Some examples follow below.

Ln 47: “Currently, 92% of the world’s population lives in areas where air pollution concentration exceeds safe limits.” WHO Guidelines for healthier air? Please quote.

Response: A reference of ‘World Health Statistics 2019: Monitoring Health for the SDGs, Sustainable Development Goals’ by WHO has now been included.

Ln 56: selection bias, please explain

Response: The selection bias refers to the bias in choosing imputation method. It was a bit confusing and now it has been elaborated as ‘… selection of imputation methods tends to result in bias’.

Ln 59: “imputation by median”: please explain imputation

Response: Imputation is the process of replacing missing data with substituted values is called imputation in statistics. The definition has now been included in the introduction part before we mention ‘imputation by median’.

Ln 92: “Furthermore, ..virtual sensor”: Why? Please explain this clearly

Response: A sentence of definition (in line 84 to 87) of virtual sensor has now included to explain the choice of this sentence.

Ln 82-83: “the deployment of this concept.. instruments”: this sentence needs to be backed up by more evidences.

Response: We have inserted a reference to show an example how a virtual sensor can be an alternative to physical sensor.

Ln 84—88: “the purpose…sensors”. You need to be more specific.

Response: Here we added ‘in terms of its accuracy and coverage’ to specify how the input-adaptive proxy can be used as a virtual sensor.

Figure 1: The procedures of developing the recursive input-adaptive proxy cannot by fully understood from the picture.

Response: More details have been supplemented in the figure. Some arrows have  also been modified for better understanding.

Ln 117: ref?

Response: The number of model created is a result of this method, therefore, there is no reference for this. To avoid misunderstanding, We removed the specification of the factorial equation.

Ln 118: “each model generated” which model? Please explain.

Response: They are the models of maximum three input variables which are created in the previous step. We have changed it to ‘each model of maximum three input variables generated in the previous step’ to provide more details in that sentence.

Ln 124: “bisquare weight function”. Please make more explicit.

Response: Equation of bisquare weight function has now been inserted in the paper. A relevant paper describing the use of bisquare weight function has also been quoted.

Ln 126: “robust fitting”. Please describe in more details.

Response: A reference has now been included in the description of robust fitting.

Ln 177: “valid data point”. Please explain.

Response: Here we meant the successful imputed data points. We revised the term into ‘output data points’ which means the same thing with less confusion.

Ln 304-305: “Since.. respectively”. Why? Please provide more details.

Response: Since the wind direction is a circular variable such that 0 degree has the same direction as 360 degrees. Now a clearer explanation and two references on which the authors used the same method have been attached.

Section 3.2 Instruments:

You could add a table for the parameters measured and their instruments.

Response: A table has now been inserted for easier understanding in a more organized way.

Ln 290: “conversion equation” please specify.

Response: The conversion equation was first introduced in the human respiratory tract model on the ICRP report by Bair in 1994. This was mentioned on the following sentence. We edited some wordings to make is clearer.

Ln 304-305: Please motivate this choice better. Why did you do this?

Response: Since the wind direction is a circular variable such that 0 degree has the same direction as 360 degrees. Now a clearer explanation and two references on which the authors used the same method have been attached.

Ln 311: “Skew distributions”: Please explain this in more details and support with references.

Response: Skewed distribution is often found in aerosols and gases variables. We also checked manually their distribution histogram to confirm it. To make it more explicit, We have now added ‘by checking the distribution histogram’ and one reference to support the statement.

Ln 336: “Thermal”, Temperature-based?

Response: In our field, the term ‘thermal seasons’ is often used. It literally means temperature-based. We have inserted ‘(temperature-based)’ to further explain the term.